# The Causal Relationship between Inflammatory Cytokines and Liver Cirrhosis in European Descent: A Bidirectional Two-Sample Mendelian Randomization Study and the First Conclusions

**DOI:** 10.3390/biomedicines12102264

**Published:** 2024-10-04

**Authors:** Shiya Shi, Yanjie Zhou, He Zhang, Yalan Zhu, Pengjun Jiang, Chengxia Xie, Tianyu Feng, Yuping Zeng, He He, Yao Luo, Jie Chen

**Affiliations:** Department of Laboratory Medicine, West China Hospital, Sichuan University, Chengdu 610041, China; shisy2000@163.com (S.S.); yanjiezhou525@163.com (Y.Z.); zhangheyq@163.com (H.Z.); zyl_wyyx@163.com (Y.Z.); jiangpengjun_scu@163.com (P.J.); chengxia_xie@163.com (C.X.); tiantianyufeng@wchscu.cn (T.F.); zypjyk@163.com (Y.Z.); hehe@wchscu.cn (H.H.); luoyao@scu.edu.cn (Y.L.)

**Keywords:** inflammatory cytokines, liver cirrhosis, mendelian randomization study

## Abstract

Background: Observational studies have highlighted the pivotal role of inflammatory cytokines in cirrhosis progression. However, the existence of a causal link between inflammatory cytokines and cirrhosis remains uncertain. In this study, we conducted a bidirectional Mendelian randomization (MR) analysis at a summarized level to illuminate the potential causal relationship between the two variables. Methods: This study utilized genetic variance in cirrhosis and inflammatory cytokines from a genome-wide association study (GWAS) of European descent. The MR-PRESSO outlier test, Cochran’s Q test, and MR-Egger regression were applied to assess outliers, heterogeneity, and pleiotropy. The inverse variance weighted method and multiple sensitivity analyses were used to evaluate causalities. Furthermore, the validation set was used for simultaneous data validation. Results: The inflammatory cytokine monocyte chemoattractant protein 3 (MCP-3) was supposedly associated with a greater risk of cirrhosis. And cirrhosis was significantly correlated with increased levels of hepatocyte growth factor (HGF). Conclusions: This study suggests that MCP-3 might be associated with the etiology of cirrhosis, while several inflammatory cytokines could potentially play a role in its downstream development. Additionally, the progression of cirrhosis was associated with elevated levels of HGF, suggesting a possible role for liver repair functions.

## 1. Introduction

Cirrhosis is a common, chronic progressive liver disease characterized by diffuse liver injury resulting from the long-term or repeated action of one or more etiologies [1]. Common causes of cirrhosis include chronic viral hepatitis, long-term alcohol abuse, fatty liver disease, autoimmune liver disease, and genetic disorders, among others [2]. Once cirrhosis develops, the rate of progression can vary from weeks to decades, but it eventually leads to liver failure with multiple complications that impair quality of life and result in death [3,4,5]. Approximately 2 million deaths from liver disease occur globally each year, 50% of which are attributed to cirrhosis [6]. Consequently, several studies have focused on the diagnosis and treatment of cirrhosis [6,7,8].

Results from observational studies suggest that inflammation is a risk factor for cirrhosis, and further evidence indicates that serum inflammatory cytokine levels are strongly associated with the severity of cirrhosis, as well as with the duration of cirrhosis and short-term mortality [9,10,11]. Although the association of inflammatory cytokines with cirrhosis and the fact that lowering inflammatory cytokine levels can inhibit the extent of liver fibrosis have been widely discussed [12], observational studies linking specific inflammatory cytokine concentrations with cirrhosis are rare and are often susceptible to multiple confounding factors; furthermore, it is unclear whether there is a causal relationship between inflammatory cytokine and cirrhosis.

The mendelian randomization (MR) study leverages large-scale datasets from genome-wide association studies (GWASs), which provide valuable insights into the associations between various risk factors and diseases. As a powerful genetic instrumental variable (IV) approach, MR uses single nucleotide polymorphisms (SNPs) as IVs to explore potential causal relationships between specific exposures and disease outcomes [13,14]. Since SNPs are naturally occurring genetic variations, their associations with diseases are not influenced by reverse causality, enabling MR to effectively mitigate the impact of unmeasured confounding factors and produce more robust and accurate causal inferences. Furthermore, by ensuring that IVs must satisfy the three assumptions of relevance, independence, and exclusion restriction, MR reduces confounding, strengthens causal inferences, and excludes the possibility of causal inversions [15,16]. In this study, we used gene variants closely associated with specific inflammatory cytokines to assess their relationship with the risk of cirrhosis. A two-sample bidirectional MR analysis was performed using GWAS data to investigate the causal link between inflammatory cytokines and cirrhosis. The workflow is shown in Figure 1. Additionally, we used the same method with two additional GWAS databases to validate our findings.

## 2. Materials and Methods

### 2.1. Data Source

There are three core assumptions in MR analysis, namely, relevance, independence, and exclusion restriction [17]. We assumed that the selected IVs are related to the risk factor but not to any confounders in the risk factor–outcome association, and that they are not connected with the outcome via any pathway other than the risk factor for interest. Here, we performed a two-sample MR analysis using summary-level GWAS data on 41 circulating inflammatory cytokines and cirrhosis to explain the causal associations. When the characteristics were similar, we selected the GWAS summary statistics with the largest sample size. The demographic data for the cohorts used are provided in Table 1.

Publicly summarized GWAS data on 41 circulating inflammatory factors were obtained from a genome-wide meta-analysis of 8293 healthy Finnish subjects [18]. The meta-analysis was based on three Finnish cohorts, including the Cardiovascular Risk in Young Finns Study (YFS) and FINRISK 1997 and 2002. The average participant ages were 37 years for the YFS and 60 years for the FINRISK survey [18].

The genetic data of cirrhosis were extracted from two available GWAS datasets. Dataset 1 on cirrhosis, excluding cirrhotic outcomes secondary to primary biliary cholangitis and primary sclerosing cholangitis, were obtained from a GWAS of 218,792 Europeans in the International Oncology Unit (IEU) Open GWAS database [19] [finn-b-CIRRHOSIS_BROAD, *n* = 218,792]. Dataset 2 (Buch S. et al.) was obtained from the United Kingdom Biobank (UKB) queue (2701 patients and 16,206 controls) [20]. Cases of cirrhosis due to primary biliary cholangitis (ICD K74.3) and primary sclerosing cholangitis (ICD K83.0) remain excluded because these autoimmune diseases target biliary (rather than hepatic) parenchyma. There was no overlap in population selection between the exposure group and the outcome group. The GWAS data used in our study were obtained with informed consent from all study participants according to the protocol approved by the respective institutional review boards; therefore, no separate ethical approval was needed for this study. The reference genome build was HG19/GRCh37. Other GWAS details were shown in the original GWAS [20].

### 2.2. Selection of IVs

The selection of IVs for cirrhosis was based on the TwoSample MR package of R 4.0.3. SNPs with *p*-values < 5 × 10^−8^ and minor allele frequencies (MAFs) > 1% were selected. These SNPs were independently genetically clumped (linkage disequilibrium R^2^ < 0.01, distance window 10,000 kb) from the European samples [21] and were not palindromic (A/T or C/G) with intermediate effect allele frequencies (40–70%). Notably, for inflammatory cytokines such as IL-16, instrumental variables with *p*-values < 5 × 10^−6^ were selected due to the small number of SNPs [21].

When the *p*-value was <5 × 10^−6^, the assumption of relevance was satisfied. To screen IVs that met the independence assumption, potential confounders were included as covariates in individual-level GWAS analyses, and multiplicative validity tests were performed. Palindromic SNPs with effector alleles and other alleles as complements were discarded. Additionally, we evaluated the strength of the association between each IV and the exposure, excluding weak IVs. We estimate the proportion of phenotypic variance explained (R^2^) and calculate F-statistics in our statistical model. The formula R^2^ = 2 × MAF × (1-MAF) × β^2^ was used to determine R^2^, where MAF represents the minor allele frequency (EAF) of the SNP, and β represents the effect of the SNP on the trait. Finally, the F-statistic for each IV was calculated using the following formula: F = R^2^ (N − 2)/(1 − R^2^) [22]. IVs with an F-statistic of less than 10 were excluded due to potential genetic confounding or measurement error. Ultimately, these rigorously selected SNPs were used as the final IVs for subsequent MR analysis.

### 2.3. Statistical Analyses

In adherence to the specified criteria for instrumental variable selection, we conducted a meticulous screening of the dataset to identify IVs that met the necessary conditions. We used the PhenoScanner database (http://www.phenoscanner.medschl.cam.ac.uk/ accessed on 14 August 2024) to search for secondary phenotypes of the selected SNPs and excluded potentially confounding SNPs.

For the MR analysis, we first performed heterogeneity and pleiotropy testing. We used the MR-PRESSO outlier test to detect and remove abnormal instrumental variables. The strength of the IVs was evaluated using F statistics, and heterogeneity and horizontal pleiotropy between inflammatory cytokines and cirrhosis were investigated by way of Cochran’s Q test. Heterogeneity was defined as a *p*-value < 0.05 indicating statistical significance. MR-Egger regression (based on intercept terms) and Mendelian randomization pleiotropy residual sum and outlier (MR-PRESSO) were used to assess horizontal pleiotropy [23,24]. When the MR-Egger outcome deviated from 0 or its *p*-value was <0.05, this indicated the presence of horizontal pleiotropy. In addition, we performed sensitivity analyses to detect and correct for pleiotropy in the estimates.

In this study, causal effect sizes were estimated mainly by way of the inverse-variance weighted method (MR-IVW method), the MR-Lasso method, the MR-Egger method, the weighted median method, the simple mode method, and the weighted mode method. When there was no heterogeneity or pleiotropy between inflammatory cytokines and cirrhosis, we performed MR analysis using the MR-IVW method, which provides accurate estimates [23]. When pleiotropy was present, analysis was performed preferentially using the MR-Egger method [25]. The weighted median method can provide valid estimates when heterogeneity and pleiotropy are present, as long as up to 50% of the instrumental variables are valid [26]. We also constructed scatter plots to visualize these univariate results.

Data analyses were performed using the MR-PRESSO (version 1.0) or TwoSample MR (version 0.5.8) packages in R software (version 4.3.2), with all tests being two-sided. Reporting follows the STROBE-MR statement.

## 3. Results

### 3.1. MR Analysis of Inflammatory Cytokines on Cirrhosis Risk

In this study, we first analyzed the training set (Dataset 1). Initially, only three cytokines had three or more genetic variants when SNPs were initially selected with *p* < 5 × 10^−8^. We then relaxed the threshold further, leading to another MR analysis (*p* < 5 × 10^−7/−6^), which identified 6 and 32 cytokines meeting our criteria, respectively. Eventually, all 41 cytokines were included with an adequate number of valid genetic variants. The variance explained by SNPs for each cytokine ranged from 1.4% to 9.6%, with all IVs statistics greater than 10, indicating robust strength. The complete data can be found in Appendix A.

A comprehensive assessment of the main analytical tools was conducted for all 41 risk factors. The IVW method was primarily used for all cytokines, except for Monocyte chemoattractant protein 3 (MCP3) and Platelet-derived growth factor BB (PDGF-BB). For MCP3, the weighted median method was selected due to significant heterogeneity in the IVW Q test, while IVW was used for PDGF-BB after excluding an outlier SNP identified using MR-PRESSO.

MR estimates of the effects of risk factors (inflammatory cytokines) on the outcome (cirrhosis) are provided in Appendix A and Figure 2A. According to the IVW method, the inflammatory cytokine MCP3 was associated with a 33% greater risk of cirrhosis (IVW-OR = 1.332, 95% CI: 1.012–1.752; *p* = 0.041), a finding consistent with the weighted median estimation method (OR = 1.518, 95% CI: 1.225–1.881; *p* < 0.001). In contrast, the inflammatory cytokine interferon gamma-induced protein 10 (IP10) (IVW-OR = 0.707, 95% CI: 0.516–0.969; *p* = 0.031) and interleukin-1 receptor antagonist protein (IL-1RA) (IVW-OR = 0.725, 95% CI: 0.548–0.959; *p* = 0.024) were associated with a reduced risk of cirrhosis. A heatmap of the MR analysis methods for the association between 41 circulating cytokine levels and the risk of cirrhosis is displayed in Figure 3A. No evidence of pleiotropy in the associations of MCP3, IP10, or IL-1RA was observed, as measured using the MR-Egger intercept (*p* = 0.090, *p* = 0.609, *p* = 0.690), and no outlier SNPs were detected using the MR-PRESSO method. Furthermore, Cochran’s Q test revealed no evidence of directional pleiotropy effects for IP10 or IL-1RA (*p* = 0.700, *p* = 0.546) (Figure 2 and Figure 3; Appendix A). Details of the SNPs are presented in Appendix A.

In the validation set (Dataset 2), bidirectional two-sample MR analysis confirmed the causal relationships. Complete data on these SNPs can be found in Appendix A. As a result, the inflammatory cytokines MCP3, IL-5, and macrophage colony-stimulating factor (MCSF) were associated with a greater risk of cirrhosis (MCP3: IVW-OR = 1.021, 95% CI: 0.988–1.054; *p* = 0.021; IL-5: IVW-OR = 1.010, 95% CI: 1.002–1.018; *p* = 0.016; MCSF: IVW-OR = 1.003, 95% CI: 1.001–1.006; *p* = 0.011), while *IP10* (IVW-OR = 0.741, 95% CI: 0.684–0.802; *p* < 0.001) was associated with a reduced risk of cirrhosis, according to the IVW method (Figure 2 and Figure 3). No evidence of pleiotropy in the associations of MCP3, IL-5, MCSF, and IP10 was observed, as measured by the MR-Egger intercept (*p* = 0.404, *p* = 0.453, *p* = 0.073, and *p* = 0.581), and no outlier SNPs were detected using the MR-PRESSO method. Furthermore, Cochran’s Q test revealed no evidence of directional heterogeneity effects for MCP3, IL-5, MCSF, or IP10 (*p* = 0.694, *p* = 0.446, *p* = 0.502, *p* = 0.981) (Appendix A; Appendix A). Details of the SNPs are presented in Appendix A.

### 3.2. MR Analysis of the Influence of Cirrhosis on Inflammatory Cytokines

When 41 inflammatory cytokines were considered as outcome factors in the reverse MR, six significant SNPs were identified as IVs for cirrhosis. The IVW method was primarily employed to evaluate the associated effects on inflammatory cytokines, as there was no evidence of heterogeneity and horizontal pleiotropy. The MR analysis revealed that cirrhosis was significantly associated with elevated levels of hepatocyte growth factor (HGF), macrophage inflammatory protein-1 beta (MIP-1β), stromal cell-derived factor-1 alpha (SDF-1α), IL-12, MCP3, and stem cell factor (SCF). Conversely, cirrhosis was linked to decreased levels of IL-2 receptor alpha (IL-2RA), IL-5, IL-7, and tumor necrosis factor beta (TNF-β) (Figure 4 and Figure 5). These findings were consistent across both the training set (Dataset 1) and the validation set (Dataset 2), with the IVW method confirming similar associations in both analyses. Detailed results and additional information are provided in the supplementary materials, including Appendix A and Appendix A.

## 4. Discussion

Inflammation, marked by abnormal cytokine levels and altered immune cell functions, is closely linked to the development of cirrhosis [9]. Cirrhosis is a common chronic progressive liver disease characterized by diffuse liver injury resulting from the long-term or repeated action of one or more etiologies [1]. Given the complex microenvironment associated with cirrhosis, understanding changes in the inflammatory landscape offers new insights into the intricate mechanisms involved. Abnormal cytokine levels indicate a systemic inflammatory state, potentially playing a key role in the progression of cirrhosis. In this study, we initially aimed to explore the casual relationships between 41 inflammatory cytokines and cirrhosis using bidirectional two-sample MR analysis. Notably, IP10 and IL-1RA were found to be associated with a reduced risk of cirrhosis. Conversely, MCP3 was linked to an increased risk, suggesting its potential role as an upstream cause of the condition. Moreover, genetically predicted cirrhosis was correlated with elevated levels of MCP3, MIP1β, SDF1α, IL-12, HGF, and SCF throughout the disease process. Interestingly, we observed a bidirectional causal relationship between higher MCP3 levels and cirrhosis. Hence, these findings suggest that persistently elevated MCP3 levels are linked to both the onset and progression of cirrhosis, highlighting the complex interplay of cytokines in liver disease pathogenesis. Several cytokines may act as initiators of liver cirrhosis, while other inflammatory cytokines are more likely to act downstream during disease progression.

Numerous studies have explored the correlation between cirrhosis and inflammatory cytokines. For instance, an observational study revealed elevated levels of proinflammatory cytokines, such as IL-6 and IL-8, in cirrhotic patients compared to healthy controls, with the levels increasing as cirrhosis progresses. Moreover, these cytokines were identified as potential predictors of liver injury and chronic liver fibrosis [1,11,27]. Salgüero et al. also noted that the inflammatory cytokines IL-6 and IL-1α correlated with the severity of cirrhosis based on the Child—Pugh score, suggesting that more severe cirrhosis is associated with higher levels of inflammatory cytokines [28]. Furthermore, key immune cells responsible for producing proinflammatory and anti-inflammatory cytokines, such as lymphocytes, monocytes, and neutrophils, exhibit abnormal functions and immune impairment in cirrhotic patients [11,29]. However, despite the wealth of observational studies, determining the causal relationship between specific inflammatory cytokines and cirrhosis remains challenging due to methodological limitations, including confounding factors and reverse causality. Consequently, elevated levels of inflammatory cytokines in cirrhotic patients may stem from the underlying cause of the disease, undetected infections, treatment side effects, or unidentified comorbidities.

The main conclusion of this MR study suggests that MCP3 may play a potential role in the progression of cirrhosis. MCP3 is a monocyte chemokine expressed by various cell types, existing as a monomer that binds to multiple receptors and attracts leukocytes, including monocytes and neutrophils [30,31]. Some studies have reported that MCP3 may be involved in liver fibrosis as well as the chronic progression of cirrhosis and hepatocellular carcinoma [32,33]. It has been observed that serum MCP3 levels in cirrhosis patients are higher than in control groups, suggesting a possible association between MCP3 levels and disease progression [33]. Additionally, there is evidence indicating that when MCP3 expression is inhibited by drugs, liver fibrosis may decrease as MCP3 levels drop [34]. Some have also proposed that the binding of MCP3 to its receptor CCR2 may promote the migration of hepatic stellate cells and Kupffer cells, potentially contributing to the progression of liver fibrosis [32]. In this study, results from both the test and validation sets indicate an association between MCP3 levels and an increased risk of cirrhosis, consistent with previous findings, though further research is still needed.

IP-10, a cytokine intricately associated with inflammation and the immune system, enhances the chemotactic activity of its receptors, thereby activating a spectrum of immune cells, including T cells, NK cells, and macrophages [35]. Anomalous expression levels of IP-10 are closely linked to a multitude of inflammatory disorders. Various studies have confirmed its significance in the etiology of liver fibrosis and cirrhosis [35,36,37]. These investigations consistently revealed heightened levels of IP-10 in patients with liver fibrosis, along with a positive correlation between IP-10 levels and the severity of fibrosis [36]. As a result, IP-10 has been proposed as a potential biomarker for assessing cirrhosis severity. However, our own research findings appear to challenge this prevailing hypothesis. According to our MR analysis, estimates from the MR-IVW method suggest that IP-10 is associated with a reduced risk of cirrhosis (OR = 0.707, 95% CI = 0.516–0.969; *p* = 0.031), and the same result was obtained in the validation set (IVW-OR = 0.741, 95% CI = 0.684–0.802; *p* < 0.001). These findings suggest that IP-10 may be associated with a reduced risk of cirrhosis. This inconsistency may be attributed to the limited dataset used in our MR analysis, which ultimately included only three SNPs associated with IP-10, potentially introducing stochastic errors. Furthermore, the specific mechanism through which IP-10 functions in the progression of fibrosis has not been determined, highlighting the imperative need for further research in this area.

When cirrhosis was defined as the exposure, the estimates from the IVW method showed that the progression of cirrhosis was significantly associated with increased levels of hepatocyte growth factor (HGF), MIP1β, SDF1α, IL-12, MCP3, and SCF and with decreased levels of IL-2RA, IL-5, IL-7, and TNF-β, which provided supportable evidence for our primary study. Among these, HGF is a protein that plays a crucial role in various biological processes, such as tissue regeneration, cell growth, and angiogenesis [38]. When the liver undergoes extensive damage and functional impairment in cirrhosis, HGF may be a critical factor in initiating the repair process. HGF has been demonstrated to promote liver regeneration by stimulating the proliferation and migration of hepatocytes [39,40,41]. Some studies suggest that increased concentrations of HGF may lead to a reduced activation of hepatic stellate cells, which are key players in the fibrosis process, thereby slowing the progression of cirrhosis [42,43,44]. Our study revealed that the progression of cirrhosis is associated with elevated levels of HGF (Dataset 1-OR = 1.052, 95% CI = 1.005–1.102; *p* = 0.031; Dataset 2-OR = 1.071, 95% CI = 1.009–1.136; *p* = 0.025), suggesting that disease progression may increase HGF expression, thereby activating liver repair functions. The mechanism of HGF in liver cirrhosis is complex, and as this is only a preliminary study, further research is needed to confirm these findings and elucidate the underlying mechanisms.

To our knowledge, this is the first MR study to investigate the causal relationship between cirrhosis and 41 inflammatory cytokines. Based on the results of our study, we suggest that MCP3 is likely correlated with the etiology of cirrhosis, while several inflammatory cytokines are more likely to be involved in the development of cirrhosis downstream. Furthermore, the progression of cirrhosis promotes an increase in HGF levels, enhancing liver repair functions. However, it is important to acknowledge several limitations. The lack of statistically significant associations for certain cytokines could be attributed to the limited sample size, which might have impacted the statistical power, or the potential influence of unaccounted-for confounding effects. Additionally, our study population primarily consists of individuals of European descent, which may introduce ancestral bias. Therefore, the findings may not be generalizable to non-European populations, and caution is needed when applying these results to other ethnic groups. We are currently recruiting cirrhosis patients to conduct genomic analysis in a Chinese cohort, although the sample size remains limited at this stage. In the future, we plan to use the recruited data to further investigate the relationship between cytokines and all-cause cirrhosis. Furthermore, our investigation utilized data from two large-scale GWASs, and the absence of specific demographic information and clinical records for the study subjects precluded subgroup analyses. It is worth noting that, although the cytokine cohort and both all-cause liver cirrhosis cohorts are of European descent, their unique backgrounds may introduce potential biases in the Mendelian randomization analysis that cannot be overlooked. Future research should carefully select datasets with more consistent population backgrounds to validate these findings and ensure the robustness of the results. We emphasize that, while we have conducted an in-depth analysis based on the existing data, future efforts should focus on collecting more comprehensive individual-level data to further validate and expand our research findings. And the results of the MR study only represent changes in risk factors across the lifespan, rather than effects at a specific time after intervention. Therefore, caution should be exercised when applying MR results to clinical interventions. Further research is needed to validate our findings and assess their relevance in clinical diagnostic processes and therapeutic decision-making.

## 5. Conclusions

In conclusion, this study suggests that MCP-3 might be associated with the etiology of cirrhosis, while several inflammatory cytokines could potentially play a role in its downstream development. Additionally, the progression of cirrhosis was associated with elevated levels of HGF, suggesting a possible role for liver repair functions. However, as this is a preliminary study, further research is needed to confirm these findings and clarify the underlying mechanisms.

## Figures and Tables

**Figure 1 biomedicines-12-02264-f001:**
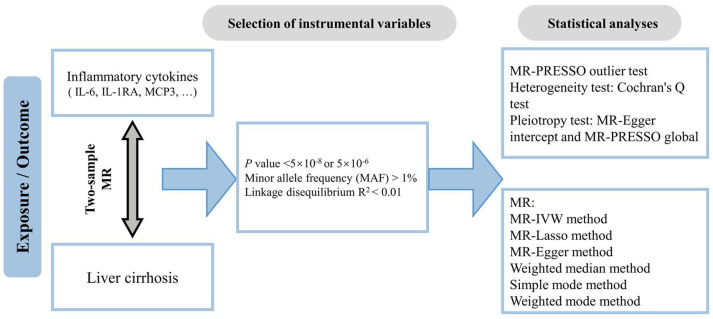
The workflow diagram of this study.

**Figure 2 biomedicines-12-02264-f002:**
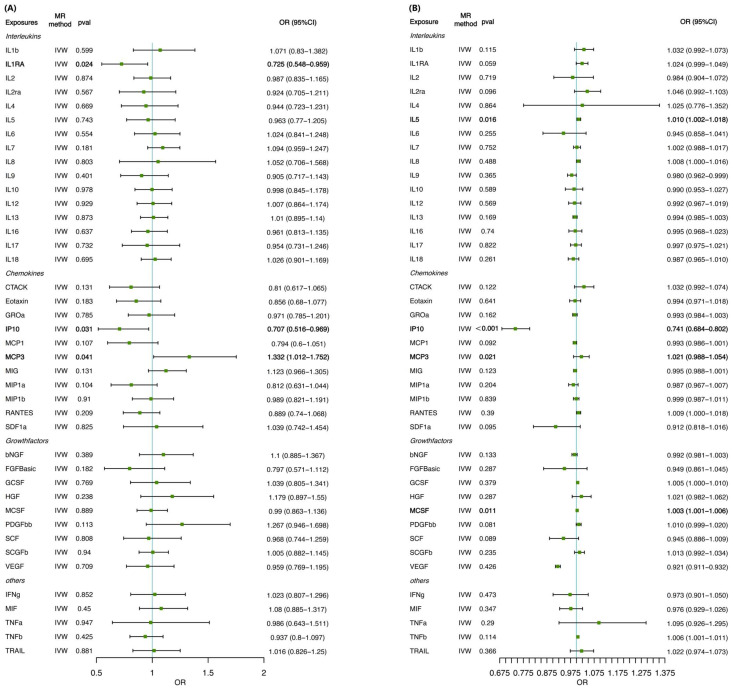
Forest plots of MR analysis of the relationship between 41 circulating cytokine levels and the risk of cirrhosis for (**A**) Dataset 1 and (**B**) Dataset 2. Bold in figure indicates *p* ≤ 0.05, which is statistically significant. CI, confidence interval; OR, odds ratio; MR, mendelian randomization.

**Figure 3 biomedicines-12-02264-f003:**
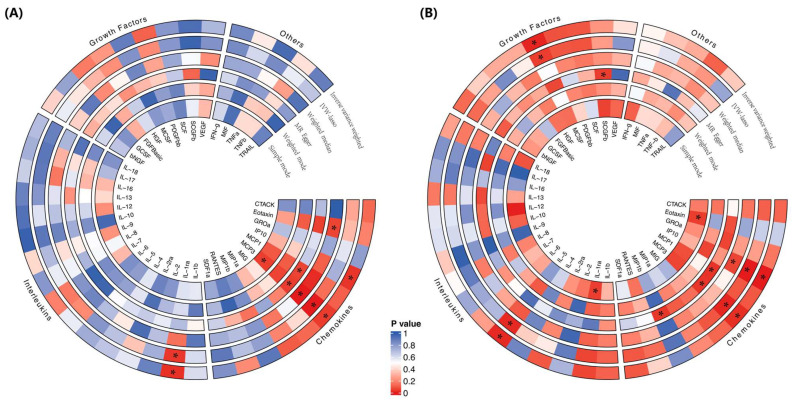
Heatmaps of the five MR analysis methods for the association between 41 circulating cytokine levels and the risk of cirrhosis for (**A**) Dataset 1 and (**B**) Dataset 2. MR, Mendelian randomization. *p* values for variance inverse variance weighting, IVW-lasso, weighted median, MR-Egger, weighted mode and simple mode are indicated from the outside in. * *p* < 0.05.

**Figure 4 biomedicines-12-02264-f004:**
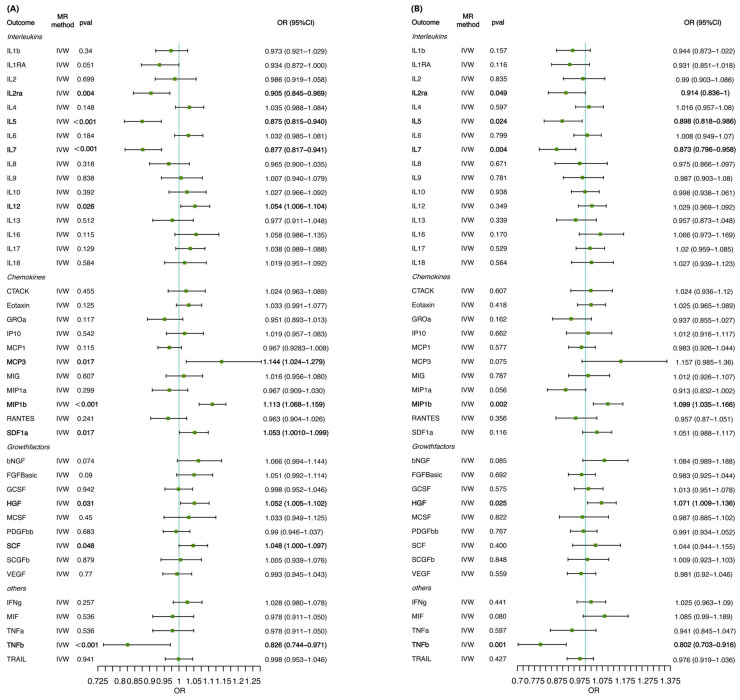
Forest plots of MR analysis of the relationship between cirrhosis and the risk of 41 circulating cytokine levels for (**A**) Dataset 1 and (**B**) Dataset 2. Bold in figure indicates *p* ≤ 0.05, which is statistically significant. CI, confidence interval; OR, odds ratio; MR, mendelian randomization.

**Figure 5 biomedicines-12-02264-f005:**
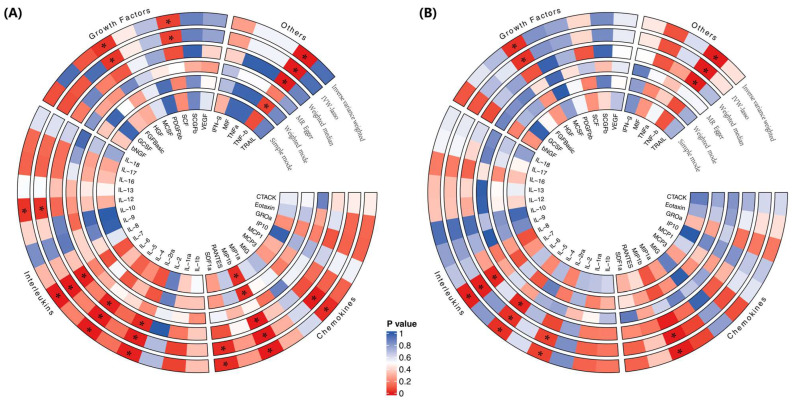
Heatmaps of the five MR analysis methods for the association between cirrhosis and the risk of 41 circulating cytokine levels for (**A**) Dataset 1 and (**B**) Dataset 2. MR, Mendelian randomization. *p* values for variance inverse variance weighting, IVW-lasso, weighted median, MR-Egger, weighted mode, and simple mode are indicated from the outside in. * *p* < 0.05.

**Table 1 biomedicines-12-02264-t001:** The demographic data for the GWAS databases used.

Characteristic		Abbreviation	GWAS ID	Sample Size	Number of SNPs	Cases Definition	Exclusion Criteria	Quality Control
Interleukins	Interleukin-1-beta	IL-1β	ebi-a-GCST004448	3309	983,642	The GWAS meta-analysis included up to 8293 Finnish individuals from three independent population cohorts: the Cardiovascular Risk in Young Finns Study (YFS), FINRISK1997, and FINRISK2002. The study cohort characteristics are reported in Appendix A in the original study. On average, the YFS participants are younger than the C23FINRISK1997 and FINRISK2002 participants (37 versus 60 years).	-	Linear regression with a probe as a dependent variable was used to test associations between cytokine-associated variants and transcripts. Age and sex were used as covariates. Genotype dosage was calculated for each included variant with Qctool (version 2) software. More details referred to the original study.
	Interleukin-1-receptor antagonist	IL-1RA	ebi-a-GCST004447	3638	9,564,741			
	Interleukin-2	IL-2	ebi-a-GCST004455	3475	9,512,914			
	Interleukin-2 receptor antagonist	IL-2RA	ebi-a-GCST004454	3677	9,583,519			
	Interleukin-4	IL-4	ebi-a-GCST004453	8124	9,786,064			
	Interleukin-5	IL-5	ebi-a-GCST004452	3364	9,450,731			
	Interleukin-6	IL-6	ebi-a-GCST004446	8189	9,790,590			
	Interleukin-7	IL-7	ebi-a-GCST004451	3409	9,692,306			
	Interleukin-8	IL-8	ebi-a-GCST004445	3526	9,517,348			
	Interleukin-9	IL-9	ebi-a-GCST004450	3634	9,567,876			
	Interleukin-10	IL-10	ebi-a-GCST004444	7681	9,793,415			
	Interleukin-12p70	IL-12p70	ebi-a-GCST004439	8270	9,799,886			
	Interleukin-13	IL-13	ebi-a-GCST004443	3557	9,539,073			
	Interleukin-16	IL-16	ebi-a-GCST004430	3483	9,551,485			
	Interleukin-17	IL-17	ebi-a-GCST004442	7760	9,786,653			
	Interleukin-18	IL-18	ebi-a-GCST004441	3636	9,785,222			
Chemokines	Cutaneous T cell attracting	CTACK	ebi-a-GCST004420	3631	9,568,408			
	Eotaxin	Eotaxin	ebi-a-GCST004460	8153	9,793,404			
	Growth-regulated protein alpha	GRPa	ebi-a-GCST004457	3505	9,528,505			
	Interferon gamma-induced protein 10	IP-10	ebi-a-GCST004440	3685	9,576,881			
	Monocyte chemoattractant protein-1	MCP1	ebi-a-GCST004438	8293	9,801,908			
	Monocyte chemoattractant protein-3	MCP3	ebi-a-GCST004437	843	7,630,881			
	Monokine induced by gamma interferon	MIG	ebi-a-GCST004435	3685	9,579,894			
	Macrophage inflammatory protein 1a	MIP1α	ebi-a-GCST004434	3522	9,519,267			
	Macrophage inflammatory protein 1b	MIP1β	ebi-a-GCST004433	8243	9,802,973			
	Regulated on activation, normal T cell expressed and secreted	RANTES	ebi-a-GCST004431	3421	9,523,827			
	Stromal-cell-derived factor 1 alpha	SDF-1α	ebi-a-GCST004427	5998	9,736,366			
Growth factors	Beta-nerve growth factor	βNGF	ebi-a-GCST004421	3531	9,537,863			
	Granulocyte-colony stimulating factor	GCSF	ebi-a-GCST004458	7904	9,788,961			
	Fibroblast growth factor basic	FGFBasic	ebi-a-GCST004459	7565	9,790,946			
	Hepatocyte growth factor	HGF	ebi-a-GCST004449	8292	9,802,538			
	Macrophage colony stimulating factor	MCSF	ebi-a-GCST004436	840	9,184,521			
	Platelet-derived growth factor BB	PDGFbb	ebi-a-GCST004432	8293	9,800,009			
	Stem cell factor	SCF	ebi-a-GCST004429	8290	9,796,683			
	Stem cell growth factor beta	SCGFβ	ebi-a-GCST004428	3682	9,574,890			
	Vascular endothelial growth factor	VEGF	ebi-a-GCST004422	7118	9,784,803			
Others	Interferon gamma	IFN -γ	ebi-a-GCST004456	7701	9,785,363			
	Macrophage Migration Inhibitory Factor	MIF	ebi-a-GCST004423	3494	9,537,573			
	Tumor necrosis factor alpha	TGFα	ebi-a-GCST004426	3454	9,500,449			
	Tumor necrosis factor beta	TGFβ	ebi-a-GCST004425	1559	6,304,298			
	TNF-related apoptosis inducing ligand	TRAIL	ebi-a-GCST004424	8186	9,698,525			
CIRRHOSIS	Cirrhosis	-	finn-b-CIRRHOSIS_BROAD	1931 cases and 216,861 controls	16,380,466	Cirrhosis ascertained through ICD-10 code K74, ICD-9 code 571 (hepatic fibrosis and cirrhosis)	The study excluded cases of cirrhosis secondary to primary biliary cholangitis and primary sclerosis cholangitis, as these autoimmune disorders are directed against the biliary (and not hepatic) parenchyma < 30 g/day for men, chronic viral hepatitis (hepatitis B and hepatitis C), autoimmune liver diseases, hereditary hemochromatosis, α1-antitrypsin deficiency, Wilson’s disease, and drug-induced liver injury	A genome-wide association study in each cohort was performed using logistic regression with adjustment for age, sex, and ten principal components of ancestry. We tested the association of fourteen million variants with a minor allele frequency of greater than 0.1% with cirrhosis in each cohort. PLINK (2015-1-25) was used for all analyses. To combine estimates across cohorts, inverse variance fixed-effects meta-analysis, as implemented by METAL (2010-9-1), was used. Quantile–quantile analysis was used to examine the presence of population stratification. No evidence of inflation was observed (lambda 1.02; Appendix A in the original study). Both additive and recessive analyses were performed.
		-	UK Biobank	2701 cases and 16,206 controls	-	Hospitalization or death due to physician-diagnosed cirrhosis: K70.2 (alcoholic fibrosis and sclerosis of the liver), K70.3 (alcoholic cirrhosis of the liver), K70.4 (alcoholic hepatic failure), K74.0 (hepatic fibrosis), K74.1 (hepatic sclerosis), K74.2 (hepatic fibrosis with hepatic sclerosis), K74.6 (other and unspecified cirrhosis of liver), K76.6 (portal hypertension), or I85 (esophageal varices). Controls were free of liver disease.	-	-

-: The information is not present in the original data.

## Data Availability

The original data presented in the study are openly available in [the International Oncology Unit (IEU) Open GWAS database] at [doi:10.1371/journal.pgen.1008629], [the United Kingdom Biobank (UKB) database] at [doi:10.1038/ng.3417] and [the University of Bristol Research Data Repository] at [doi:10.1016/j.ajhg.2016.11.007], and no new data were generated in this research.

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
