# Peer review of "The Causal Relationship between Inflammatory Cytokines and Liver Cirrhosis in European Descent: A Bidirectional Two-Sample Mendelian Randomization Study and the First Conclusions"

_biomedicines, 2024, doi:10.3390/biomedicines12102264_

Round 1

Reviewer 1 Report

Comments and Suggestions for Authors

Dear Authors! My congratulations on such an interesting scientific approach to the problem of liver cirrhosis. I propose to supplement: the end of the title of the article: "... and the first conclusions"; in 'Keywords' instead of 'cirrhosis' write 'liver cirrhosis'. Let me ask you a few questions. 1) Did you take into account the following factors when assessing heterogeneity in both samples: a) in the Finnish cohort of healthy individuals (ref. 18), the 218,792 Europeans (ref. 19) from the GWAS database and the 2,701 UK residents (ref. 20) with liver cirrhosis have their own historical patterns of alcohol consumption; b) the proportion of ethnic Europeans and migrants and viral and alcoholic cirrhosis in the Finnish, pan-European and UK cohorts. 2) In the 'Discussion' you wrote about the lack of 'specific demographic information in the two large-scale GWAS studies'. What is this information and how could it affect your conclusions. In the 'Discussion' paragraph about MCP3 (ref. 33) you write about the role of the latter in fibrosis and cirrhosis of the liver, and reference 33 is devoted to systemic sclerosis. 4) In the 'Discussion' and 'Conclusions' you are the first to statistically substantiate the leading role of MCP3 in the etiology of liver cirrhosis based on the results of Mendelian randomization, but you provide only two references to clinical (34,35) and one (36) experimental studies to confirm your hypothesis. 5) Was there a need for MR-STROBE and MR-Lasso, as a variant of IVW analysis, to enhance the reliability of your data, especially since in the 'Discussion' you write about the "limited sample size".

Reviewer 2 Report

Comments and Suggestions for Authors

Dear Author,

The manuscript submitted for reviewed revealed a remarkable scientific work. The author in the current work is trying to highlight the role of inflammatory cytokines in cirrhosis progression. He performed a bidirectional Mendelian randomization analysis to illuminate the potential causal relationship between the two variables. The author utilized genetic variance in cirrhosis and inflammatory cytokines from genome-wide association study. The MR-PRESSO outlier test, Cochran's Q test, and MR-Egger regression were used to estimate outliers, heterogeneity, and pleiotropy. The inverse variance weighted method, along with multiple sensitivity analyses, was used to evaluate causalities.

However, some comments and recommendations which, when appropriately addressed, may enhance the quality of the paper.

First: The inclusion and exclusion criteria should be clear.

Second: Author did not demonstrate how he can eliminate the intrinsic uncertainty in the instrumental variable assumptions? which means that there remains somewhat uncertainty in the putative causal conclusion!

Third: The author needs to explain the lack of statistically significant associations for some inflammatory cytokines and highlight that point!

Fourth: Regarding quality assessment, what are the appropriate measures that taken to avoid the risk of bias?

Fifth: The Author didn’t define all outcomes for which data were sought.

Sixth: The present study population composed of individuals of European ancestry. This issue may cause potential ancestry bias! What about the populations of non-European ancestry?!

Seventh: Mendelian randomization analyses is only a statistical method that uses genetic associations with both exposure and the outcome to infer possible causality. The MR findings reflect differences across the life course in a risk factor, not the effects at a specific time after an intervention is implemented. So, It should be treated with caution when applied to clinical intervention. The author should highlight that point.

Eighth: The participants of the datasets included here are all of European descent, So, the author cannot generalize the statistical causal associations to other ethnicities and races?!

Ninth: Some detailed information of each individual is missing. So, the author should clarify how the demographic distributions of disease GWAS match well with the IDP GWAS?! Explain!

Tenth: There are some linguistic and grammatical errors that should be rephrased and written in a correct style.

Good luck

Comments on the Quality of English Language

Minor editing of English language required

Reviewer 3 Report

Comments and Suggestions for Authors

The subject of cirrhosis in liver and its progression is very important thus the knowledge regarding mechanisms mamy be of high significance. However, I Think that Authors jump to conclusions based on very limited data. No other factors Except genetics were taken under consideration. As tey themselves pointed, studied group co consisted of European population therefore conclusions are too general. I suggest to rewrite the part concerning conclusion and stressed that it is rather preeliminary study.

Comments on the Quality of English Language

English is generally good quality 

Round 2

Reviewer 2 Report

Comments and Suggestions for Authors

Dear Author ,

Good work! However, Your feedback on the comments 6,7,8,and 9 need to be completed and you should be clear in your response. Please, answer all the comments one by one in an appropriate way. I am looking forward to receiving it.

Good Luck

Comments on the Quality of English Language

Minor editing of English language required.

Round 3

Reviewer 2 Report

Comments and Suggestions for Authors

Dear Author,

After reviewing your manuscript again, I think your manuscript looks good now. You modified and corrected all mistakes. Good work!

Regards